# Poor glycaemic control contributes to a shift towards prothrombotic and antifibrinolytic state in pregnant women with type 1 diabetes mellitus

**Maciej Osiński**  *, **Urszula Mantaj, Małgorzata Kędzia, Paweł Gutaj, Ewa Wender-Ożegowska**

Department of Reproduction, Chair of Obstetrics, Gynaecology and Oncology, Poznan University of Medical Sciences, Poznan, Poland

* maciejosinski@me.com

**Data Availability Statement:** All relevant data are within the paper and its Supporting Information files.

## Abstract

### Ojectives

Thrombotic and antifibrinolytic influence of Diabetes mellitus type 1 (T1DM) on haemostasis have been well demonstrated. There has been no research assessing the influence of poor glycemic control on thrombus formation under flow conditions in vitro or in pregnant type 1 diabetic women to date.

### Patients/Methods

This study compared singleton pregnant T1DM women (n = 21) and control pregnant subjects without any metabolic disease (n = 15). The T1DM group was divided into two subgroups of sufficient (SGC-DM; $Hb_{A1c} \leq 6,5\%$, n = 15) and poor glycaemic control (PGC-DM; $Hb_{A1c} > 6,5\%$, n = 6). Evaluation of the whole blood thrombogenicity we assessed using T-TAS® at a shear rate of 240 s-1 (Total-Thrombus Analysis System, Zacros, Japan).

### Results

Blood clot formation initiation time (T10) was significantly shortened in PGC-DM subgroup when compared to SGC-DM subgroup (p = 0,03). The area under the curve (AUC30) of blood clot time formation and the MPV (mean platelet volume) values were substantially higher in the PGC-DM subgroup in comparison to the SGC-DM group (p = 0,03). Negative correlations were noted between $Hb_{A1c}$ and T10 values (p = 0,02) and between T10 and MPV values in the T1DM group (p = 0,04).

### Conclusions

Poor glycaemic control in T1DM subjects triggers a shift towards a prothrombotic and antifibrinolytic state. This phenomenon can be detected using the novel system for quantitative assessment of the platelet thrombus formation process under flow conditions in vitro. The alteration of T-TAS values in PGC-DM subgroup proves that a poor glycemic control-related

**Funding:** The authors received no specific funding for this work.

**Competing interests:** The authors have declared that no competing interests exist.

shift of the equilibrium toward thrombogenesis occurs in this group of patients. Our findings need a further elucidation in research on more massive data sets to be confirmed.

## Introduction

Diabetes mellitus (DM) involves a group of chronic metabolic disorders with existing chronic hyperglycemia, defects of insulin action, secretion or both; with altered fat, protein and carbohydrate metabolism [1]. Diabetes is increasingly affecting the world's population at an alarming rate. It is estimated that DM will affect 500 million people worldwide by 2035 [2]. Since diabetes requires lifelong care and treatment, the urgent need to improve our understanding of its pathophysiology and complications arises.

DM is a major risk factor for cardiovascular disease [3]. As mortality related to vascular disease such as coronary heart disease (CHD) in the general population decreases, conversely it increases in diabetic patients [2]. It is associated with metabolic alterations such as hyperglycemia, dyslipidemia, hypertension and hypercoagulability [4]. Diabetes is associated with macrovascular as well as with microvascular complications. Molecular mechanisms linked to increased thrombogenicity with Diabetes are still elusive and poorly understood. The existence of hypercoagulability in type 1 and type 2 diabetes mellitus has been known for a few decades now [5]. The hypercoagulability may be related to poor glycemic control and that the hemostatic disturbances precede demonstrable vascular complications [6]. Numerous factors are triggering thrombogenicity in diabetic patients. Hyperglycemia and chronic hyperinsulinemia in type 2 diabetes mellitus (T2DM), or related to exogenous insulin overdose in T1DM, play an essential role in haemostasis alteration found in DM. Abnormalities in all phases of coagulation occurring in those patients include platelet hyperreactivity, impaired fibrinolysis, endothelial dysfunction, leukocyte activation, chronic low-grade inflammation, increased serum coagulant factors concentration and microparticle involvement [2].

In DM serum concentrations of von Willebrand Factor (vWF), clotting factor VII and fibrinogen increase, while antithrombin, endothelial thrombomodulin and protein C concentrations decrease. Moreover, endothelium monocytes and platelets demonstrate increased microparticle production [7,8].

In diabetic patients, platelet dysfunction and hyperreactivity have been demonstrated [7,8]. Processes of platelet adhesion, activation and aggregation in these groups of patients are augmented [9,10].

In Diabetes, platelet dysfunction occurs due to intracellular hyperglycemia related to insulin-independent glucose transporters in the platelet cell membrane [11]. It contributes to an increase of intracellular calcium concentration. As a corollary, it promotes the process of platelet granule degranulation [12]. Hyperglycemia increases P-selectin expression, affects serum osmotic values, promotes non-enzymatic glycosylation process and triggers activation of PKC (protein kinase C) [2]. Surface platelet proteins glycoxidation result in membrane fluidity, decrease and eventually increased platelet adhesion [13]. Defects of insulin action alter platelet response to antiplatelet drugs [12,14]. In DM patients it is observed that there is an increase in GpIIb/IIIa expression and amplification of ADP/P2Y12 signaling and there is also an increase in the production of derivatives of oxygen (ROS), which contributes to platelet dysfunction [15].

Platelet size is an essential marker of platelet function. The mean platelet volume (MPV) remains an indicator of platelet activation. Thus, MPV is also considered as an important

biological variable predicting risk of DM vascular complications. Liu et. al showed an association of MPV with the severity of diabetic retinopathy [16]. It has been evidenced that in DM subjects platelet hyperreactivity may be observed at the stage of megakaryopoiesis. Moreover, activation of leukocytes occurring in DM promotes interaction of those cells with endothelium and formation of leukocyte-platelet aggregates [7]. In DM patients process of platelet-dependent thrombin formation is augmented and thus plasma fibrinopeptide A concentration is higher in diabetic subjects [17]. Moreover, there is a substantial body of evidence that fibrinolysis decreases in DM patients. However this process is only partially understood [18,19]. It is evidenced that the main factors triggering hypofibrinolysis in DM are altered fibrin structure and impairment of the fibrinolytic system. The structure of fibrin clots in DM is more compact, thus more resistant to fibrinolysis. Elevated levels of thrombin and both qualitative and quantitative alteration in fibrinogen cause probably this phenomenon. Impaired fibrinolysis occurs due to elevated levels of plasminogen activator inhibitor (PAI-1), glycation of plasminogen and antifibrinolytic proteins incorporation into fibrin clots such as complement C3 and plasmin inhibitor [20].

In a Danish nationwide cohort study, Fangel et. al demonstrated that type 1 and type 2 DM are associated with a similar risk of a thrombotic event [21]. Fangel et al. evidenced on the basis of a large cohort (n = 5386) that VTE episode risk in DM patients with $Hb_{A1c}$ levels 49–58 mmol/mol (6,6–7,5%) and with $Hb_{A1c}$ levels more than 58 mmol/mol (7,5%) are 1,49 and 1,59 fold increased respectively in comparison to DM subjects characterized by $Hb_{A1c}$ levels $\leq$ 48 mmol/mol ($\leq$6,5%) [22]. Neergaard-Petersen et. al. showed a correlation of increased levels of $Hb_{A1c}$ with increased platelet aggregation [23].

Pregnancy itself predisposes to a prothrombotic state. In pregnancy, the serum concentration of prothrombotic agents increases (clotting factors VII, VIII, X, von Willebrand Factor, fibrinogen, plasminogen activator inhibitor type 1 (PAI-1) while the concentration of anticoagulants, such as free protein S, is decreased [7,24].

There are numerous methods for assessing haemostasis, however, screening coagulation assays are insensitive to detect hypercoagulation adequately and thus remain unable to evaluate thrombotic risk.[7] Efforts have been made to use global, integral assays to mimic and reflect aspects of haemostasis *in vitro*. An example of such is the Total Thrombus-formation Analysis System (T-TAS)[R]. This system enables the evaluation of the *in vitro* thrombus formation process under flow conditions. TTAS is an innovative microchip flow-chamber system enabling quantitative assessment of the platelet thrombus formation process. This assay represents noteworthy improvements over conventional platelet function assays [25].

While the thrombotic and antifibrinolytic influence of DM on haemostasis has been well demonstrated in many previous studies, conventional coagulation assays remain insensitive to detect hypercoagulation and to evaluate the risk of developing thrombosis. To date, there has been no research assessing the influence of poor glycemic control in DM subjects on the process of thrombus formation under flow conditions *in vitro* and no such data has been obtained in diabetic pregnant women.

Our study aimed to evaluate if poorly controlled Type 1 Diabetes increases the risk of changes in haemostasis in diabetic pregnant women and whether the T-TAS enables the evaluation of these changes with higher sensitivity and predictivity than the standard methods.

## Materials and methods

### Subjects

We collected data of patients who were referred to the tertiary unit for diabetic pregnant women in the Division of Reproduction, Poznan University of Medical Sciences, in 2017–

 

2018. The analysed groups of women compared singleton type 1 diabetic women in the first and early-2nd trimester of pregnancy (n = 21) with a comparable group of non-diabetic, healthy pregnant women, matched with: age, BMI, blood pressure, week of gestation (n = 15), hospitalized in our department. The diabetic group was divided into subgroups of sufficient (SGC-DM, n = 15) and poor glycemic control (PGC-DM, n = 6) according to American Diabetes Association and Polish Society of Gynaecologists and Obstetricians guidelines. They both recommend a sustained level of $Hb_{A1c}$ values < 6.5% in preconception counselling [26,27]. Sufficient glycemic control group was characterized by $Hb_{A1c}$ values ≤6,5%, (48 mmol/mol) and poor glycemic control by $Hb_{A1c}$ values > 6,5%, (n = 6). In both groups the same exclusion criteria were used: a prior episode of venous or arterial thrombosis, diagnosed acquired or inherited thrombophilia, neoplasm and uncertain family history of thrombosis, multiple pregnancy and hypertension.

## Methods

The Ethics Committee approved the study protocol of Poznan University of Medical Sciences (approval number 300/17, 2/3/2017). We took informed written consent from all patients.

We collected blood samples for standard conventional laboratory tests and for T-TAS at the same time on a fasting state from the antecubital veins. All biochemical tests were performed in the accredited laboratory of the academic hospital, holding certificates of quality management ISO 9000.

Evaluation of whole blood thrombogenicity—thrombus formation under flow condition was assessed immediately after taking the blood, using T-TAS® at a shear rate of 240 s⁻¹ (Total Thrombus Analysis System, Fujimori Kogyo, Zacros, Japan, AR-chip) equipped with AR microchip and thrombogenic surfaces (collagen with thromboplastin). For each test, patient blood samples were analysed for thrombus formation area under the flow pressure curve for 30 min (AUC30 indicates area under the flow pressure curve for the 30 min.), time of blood clot formation initiation (T10) and OT or T80 (an occlusion time—time of complete thrombus formation inside the AR-chip). T80-T10 –time of growth of the thrombus on the AR chip, (time between time of blood clot formation initiation and occlusion time). T10 is defined as the time of the onset of a thrombus formation. It resembles the duration of the flow pressure from baseline values to the 10 kPa pressure values. It is related with a partial occlusion of microcapillary. T80 (OT) represents time of a complete occlusion of the microcappilary, occurring with a pressure 80kPa. T80-T10 is the interval between T10 and OT or T80 and represents the rate of growth of a thrombus. AUC30 remains an area under the 80 kPa flow pressure curve for 30 min after the start of assay in AR chip.

Statistical analysis was performed using PQ STAT® statistical software. Data was checked for normality using the Shapiro-Wilk test and then appropriate parametric or nonparametric post-hoc tests were used to assess differences between the parameters across the three groups. For nonparametric ANOVA Kruskal-Wallis, Dunn-Bonferroni post-hoc tests were used and for parametric One Way ANOVA, Fisher-LSD post hoc were used. To assess differences across two groups U Mann-Whitney test was used. (There were only such 2 variables, both nonparametric), variables values are presented as mean ± 1 standard deviation or median (interquartile range) appropriately to use of nonparametric and parametric, respectively. P-value <0.05 was considered as statistically significant, with a Bonferroni correction used for a multiple group testing. Pearson's and Spearman's correlation rank statistical tests were used.

## Results

In this particular study, we aimed to compare a whole blood thrombus formation process under flow conditions in diabetes mellitus type 1 and non-diabetic pregnant individuals. Besides, we performed conventional standard laboratory investigations in all groups.

### Clinical characteristics and standard laboratory investigations

Comparable groups of patients were matched in basic demographic and clinical characteristics. There were no differences in basic variables depicting patients' age, BMI, systolic and diastolic blood pressure, week of gestation, serum values of CRP, ALT, AspAT, lipids, creatinine, electrolytes, APTT, PT, antithrombin, fibrinogen, INR, PLT, LEU, HGB, HCT and other numerous laboratory investigations listed below. In diabetic patients, there were no differences in the age of the disease diagnosis and known duration of the disease. Clinical characteristics and the most important data are shown in Table 1. All variables obtained in the study are presented in S1 Table.

The T1DM group was divided into subgroups of sufficient glycaemic control (SGC DM, n = 15) and poor glycaemic control (PGC DM, n = 6). Mean $Hb_{A1c}$ values were statistically significantly different (p<0.000001, Fig 1) as were the fasting blood glucose levels between those two diabetic groups (p = 0,002). Mean MPV value was significantly higher in the PGC DM group in comparison to the SGC DM group (p = 0,03, Fig 2). We found differences in D-dimers concentration between the groups. D-dimers concentration was significantly lower in the PGC DM group in comparison to the SGC DM group (p = 0,01, Fig 3). There were no differences between the sufficient glycemic control DM group and the healthy pregnant group (p = 0,49).

### Total thrombus analysis system results

We observed differences between diabetic groups in T10, AUC30 values (T-TAS) and we found relations between T-TAS findings and other metabolic parameters: $Hb_{A1c}$, fasting blood glucose, MPV and D-dimers (p<0,05). T10 values were significantly shortened in the poor glycaemic control DM group (p<0,05, Fig 4), but there was no difference in T10 values between healthy pregnant women and the sufficient glycemic control DM group (p = 0,77). AUC values were significantly higher in the poor glycemic DM group (p<0,05, Fig 5). We found a significant correlation between $Hb_{A1c}$ cconcentration and T10 values (p = 0,02). However $Hb_{A1c}$ and MPV, as well as T10 and D-dimers values didn't present a significant correlation (p>0.05). Besides, we found a significant correlation between $Hb_{A1c}$ and D-dimers values (p = 0,0009). We also have found a negative correlation between the MPV and T10 values in the entire diabetic group (p<0.05).

## Discussion

In the present study, we investigated the prothrombotic influence of poor glycemic control in pregnant women suffering from type 1 diabetes mellitus using a novel automated microchip flow chamber system for the quantitative analysis of the thrombus formation Total Thrombus-formation Analysis System (T-TAS®). T-TAS enables the assessment of the whole blood thrombogenicity [25]. We designed this study to evaluate various antiplatelet drugs prophylaxis against thrombosis, based on the phenomenon that platelets remain crucial to initiate thrombus formation. This system further allows an evaluation of *in vitro*

**Table 1. Characteristics of the studied groups.**

| Variable | A CONTROL NON-DM GROUP* | B SUFFICIENT GLYCAEMIC CONTROL T1DM GROUP * | C POOR GLYCAEMIC CONTROL T1DM GROUP * | p** | A vs B p*** | A vs C p*** | B vs C p*** |
|---|---|---|---|---|---|---|---|
| | n = 15 | n = 15 | n = 6 | | | | |
| Clinical characteristics | | | | | | | |
| Age [years] | 29,1 ± 4,6 | 28,67 ± 4,6 | 24,2 ± 3,7 | 0,08 | 0,81 | 0,03 | 0,05 |
| Body mass index, [kg/m2] | 22,8 (20,6–25,2) | 23,8 (22,3–24,9) | 25,2 (21,0–28,0) | 0,66 | 1,00 | 1,00 | 1,00 |
| Systolic Blood Pressure [mmHg] | 105,9 ± 12,0 | 108,6 ±10,1 | 101,7 ± 14,2 | 0,55 | 0,53 | 0,45 | 0,23 |
| Diastolic Blood Pressure [mmHg] | 66,5 ± 8,3 | 70,7 ± 8,7 | 61,3 ± 6,3 | 0,05 | 0,15 | 0,20 | 0,02 |
| Known duration of the disease [years] | NA | 12,00 (6–17,5) | 9,00 (6,3–12,5) | 0,37**** | - | - | 0,37 |
| Age of T1DM diagnosis [years] | NA | 17,1 ± 7,6 | 15,8 ± 6,2 | 0,73***** | - | - | 0,73 |
| Week of gestation | 12,00 (12–13,5) | 12,00 (11,0–13,0) | 11,00 (9,5–14,0) | 0,31 | 0,85 | 0,48 | 1,00 |
| Comparative T-TAS findings | | | | | | | |
| **T10 [s]** | **395,1 ± 158,4** | **419,3 ± 154,9** | **224,7 ± 34,1** | **0,03** | **0,77** | **0,01** | **0,01** |
| **AUC30** | **1804,0 (1738,4–1939,1)** | **1814,7 (1628,9–1906,7)** | **2024,0 (2003,2–2035,5)** | **0,03** | **1,00** | **0,09** | **0,03** |
| OT [s] | 528,0 (411,0–582,0) | 510,0 (437,5–653) | 329,0 (328,3–355,3) | 0,09 | 1,00 | 0,25 | 0,08 |
| T80-10 [s] | 134,0 (99,5–140,5) | 124,0 (105–158,5) | 104,0 (93,5–117,5) | 0,54 | 1,00 | 1,00 | 0,80 |
| Comparative Standard laboratory investigations | | | | | | | |
| **HBA1C [%]** | **4,8 ± 0,4** | **5,8 ± 0,6** | **7,9 ± 0,8** | **<0.000001** | **0,00001** | **<0.000001** | **<0.000001** |
| **Fasting Glucose, [mg %]** | **78,3 (99,5–140,5)** | **84,2 (54,4–92,9)** | **120,3 (114,2–147,1)** | **0,00** | **1,00** | **0,002** | **0,01** |
| CRP [mg/L] | 3,7 (2,0–7,7) | 1,6 (1,3–2,8) | 3,4 (1,8–15,5) | 0,17 | 0,24 | 1,00 | 0,59 |
| **MPV [fl]** | **10,9 ± 0,8** | **10,4 ± 0,7** | **11,1 ± 0,5** | **0,04** | **0,04** | **0,48** | **0,03** |
| ALT [U/L] | 11,2 (8,3–12,5) | 10,4 (9,7–12,2) | 9,7(8,4–10,2) | 0,34 | 1,00 | 0,58 | 0,47 |
| AST [U/L] | 14,5 ± 3,1 | 14,0 ± 2,9 | 12,7 ± 3,2 | 0,49 | 0,62 | 0,24 | 0,41 |
| Total cholesterol [mg/dL] | 187, ± 27,0 | 165,6 ± 34,9 | 162,0 ± 7,5 | 0,08 | 0,05 | 0,08 | 0,80 |
| HDL-C, [mg/dL] | 81,8 (64,1–97,0) | 78,20 (66,6–89,7) | 77,4 (65,1–89,8) | 0,84 | 1,00 | 1,00 | 1,00 |
| LDL-C, [mg/dL] | 80,0 ± 34,1 | 70,4 ± 28,3 | 61,4 ± 14,7 | 0,39 | 0,37 | 0,20 | 0,53 |
| Triglycerides, [mg/dL] | 92,6 (75,7–127,7) | 70,7 (59,9–88,1) | 115,95 (65,0–164,2) | 0,24 | 0,35 | 1,00 | 0,69 |
| Serum creatinine, [mg/dL] | 0,5 (0,5–0,6) | 0,6 (0,5–0,6) | 0,6 (0,5–0,6) | 0,82 | 1,00 | 1,00 | 1,00 |
| Antithrombin, [%] | 88,8 ± 7,5 | 85,9 ± 11,8 | 93,3 ± 5,8 | 0,27 | 0,58 | 0,24 | 0,12 |
| APTT [s] | 26,6 (25,6–28,6) | 29,2 (27,5–30,5) | 28,7 (26,6–32,9) | 0,11 | 0,13 | 0,49 | 1,00 |
| **D-dimers [ng/mL]** | **577,0 (425,0–999,5)** | **466,00 (359,0–585,5)** | **231,5 (191,0–246,5)** | **0,0004** | **0,46** | **0,0002** | **0,01** |
| Fibrinogen [g/L] | 4,1 ± 0,9 | 4,5 ± 0,8 | 4,6 ± 2,0 | 0,57 | 0,27 | 0,31 | 0,86 |
| PT [s] | 11,3 (11,0–11,4) | 11,3 (11,0–11,9) | 11,2 (10,9–11,7) | 0,71 | 1,00 | 1,00 | 1,00 |
| INR | 1,0 (1,0–1,0) | 1,0 (1,0–1,1) | 1,0 (1,0–1,1) | 0,71 | 1,00 | 1,00 | 1,00 |
| Prothrombin index [%] | 97,3 (96,5–100,0) | 97,3 (92,8–100,5) | 98,3 (94,4–100,9) | 0,71 | 1,00 | 1,00 | 1,00 |
| Leucocytes [G/L] | 8,3 (7,4–9,9) | 7,9 (7,1–9,0) | 8,8 (7,9–9,2) | 0,61 | 1,00 | 1,00 | 1,00 |
| Erythrocytes [T/L] | 4,2 ± 0,2 | 4,3 ± 0,4 | 4,3 ± 0,3 | 0,48 | 0,32 | 0,74 | 0,68 |
| Platelet count [G/L] | 223,8 ± 44,5 | 248,7 ± 56,6 | 274,2 ± 82,6 | 0,18 | 0,24 | 0,08 | 0,36 |
| HGB [mmol/L] | 7,7 ± 0,5 | 7,7 ± 0,5 | 7,8 ± 0,4 | 0,93 | 0,94 | 0,83 | 0,79 |
| HCT [L/L] | 0,4 ± 0,02 | 0,4 ± 0,02 | 0,4 ± 0,02 | 0,75 | 0,50 | 0,57 | 0,95 |

*(Continued)*

**Table 1.** (Continued)

| Variable | A CONTROL NON-DM GROUP* | B SUFFICIENT GLYCAEMIC CONTROL T1DM GROUP * | C POOR GLYCAEMIC CONTROL T1DM GROUP * | p** | A vs B p*** | A vs C p*** | B vs C p*** |
|---|---|---|---|---|---|---|---|
| | n = 15 | n = 15 | n = 6 | | | | |
| MCV [fL] | 85,1 ± 4,3 | 83,8 ± 6,1 | 84,8 ± 2,5 | 0,77 | 0,46 | 0,90 | 0,67 |

* Mean with SD (standard deprivation) are presented for parametric data. Median with 1st and 3rd quartile are reported for non-parametric data.

** For parametric: One Way ANOVA and for nonparametric: ANOVA Kruskal-Wallis tests were used.

*** Fisher LSD and Dunn-Bonferroni post-HOC tests were used for parametric and non-parametric data, respectively.

**** U Mann-Whitney test.

***** t-student.

NA—not applicable.

thrombus formation process under flow conditions. It may be also used to assess clot formation in haemostasis alteration such as factor VIII deficiency (mouse model) [28]. Previous research on T-TAS showed that the system may be useful in assessing antithrombotic agents effects [25,29]. So far, the impact of poor glycemic control in diabetic pregnancies on haemostasis using T-TAS® and the impact of diabetes in pregnancy itself on this process had not been analysed yet.

We have documented that poor glycemic control in diabetic pregnancies is associated with increased thrombogenicity. It is also known that augmented blood clotting in those patients could not be recognizable by using standard clotting test such as the prothrombin time and activated partial thrombin time (APTT).

Hence our data strongly suggests that T-TAS might be a sensitive diagnostic tool to detect a prothrombotic state, especially in high-risk patients predisposed to develop thrombosis, such as pregnant women with poorly controlled type 1 diabetes mellitus.

In our study, we have also found significantly higher MPV in the poor glycemic control DM subjects. Our results are consistent with previous research published by Colak et al. who established that MPV values were higher in gestational diabetes (GDM) in comparison with a non-GDM control group. They concluded that MPV can be used to predict GDM in the first trimester of pregnancy [30]. Coban et al. had also found suggestively higher MPV values in obese patients [31]. As mentioned in the introduction, it has been shown that MPV remains an indicator of platelet activation. Consequently, MPV is considered as an important biological variable predicting risk of DM vascular complications, which is equivalent to insulin resistance and with the prothrombotic state.

A mean platelet volume increase is associated with increased insulin resistance [32]. A study conducted by Varol et al. had shown a linkage between MPV values and insulin resistance. In this study, an upsurge in MPV values had been observed in insulin-resistant non-obese, non-diabetic CAD (coronary artery disease) patients in comparison to insulin-sensitive non-obese, non-diabetic CAD patients [33]. Radha and Selvam had established increased MPV values in uncontrolled DM patients, and a more frequent occurrence of DM complications in this group of patients [34]. We found a negative Pearson correlation between MPV and T10 in DM patients, which remains consistent with previously cited research.

There is a considerable body of evidence that increased MPV values are associated with increased risk of an ischemic vascular episode such as myocardial infarction [35,36]. Increasing MPV may serve as a predictor for VTE, particularly VTE of unprovoked origin [37].

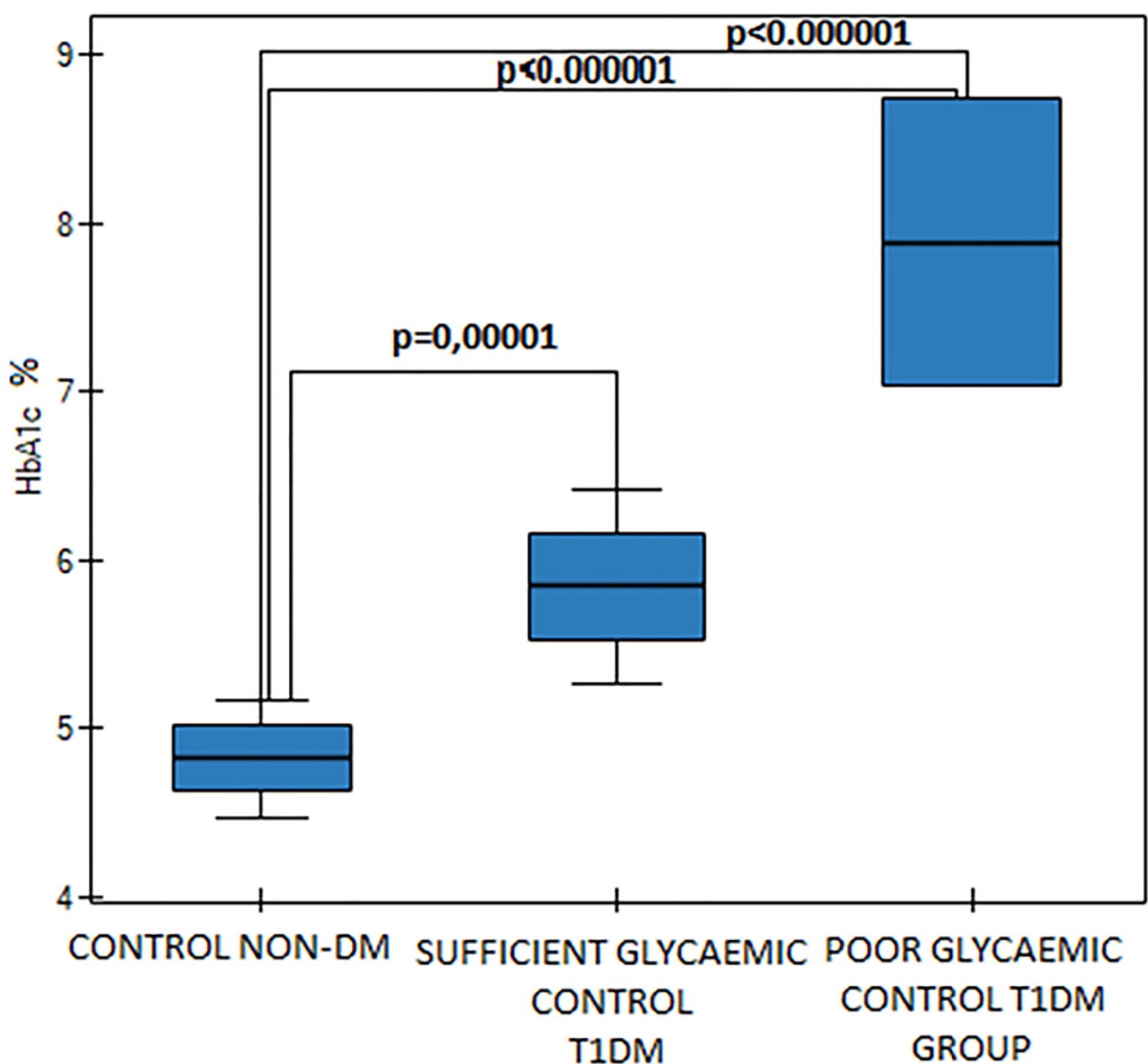

**Fig 1. Hb$_{A1c}$ values in the control group and in the sufficient and the poor glycaemic control DM groups, p<0.000001 (One Way ANOVA, Fisher LSD).**

Conversely, it has been shown that a low MPV remains an independent risk factor for increased blood loss during surgery, such as radical hysterectomy [38].

In our study, the duration of Diabetes was not different in the compared SCG and PGC diabetic groups. Thus the length of the disease does not appear to be the main factor that differentiates these groups in relation to diabetic complications. A study based on a nationwide Danish registry revealed that in patients with atrial fibrillation, a longer duration of diabetes

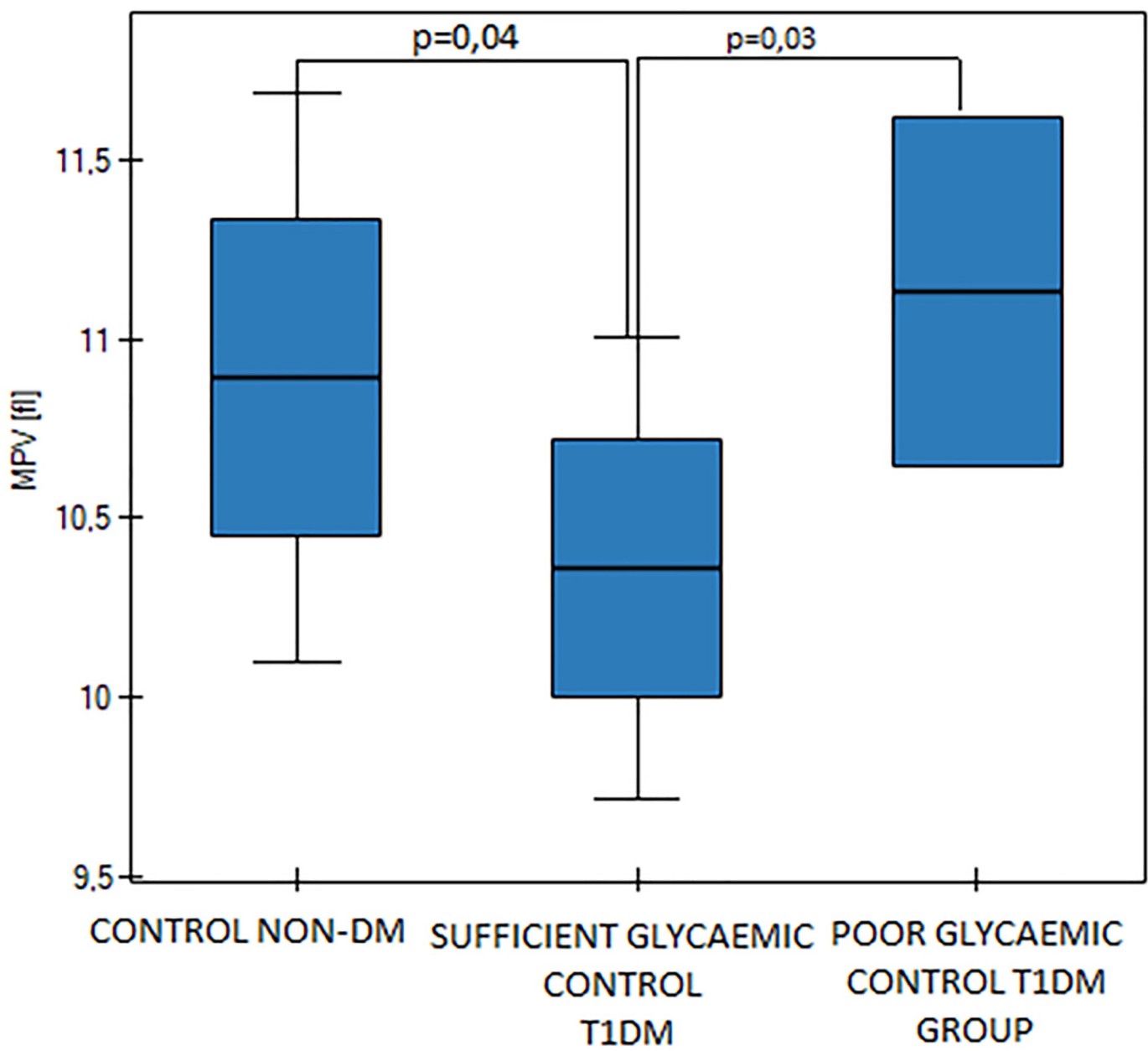

**Fig 2. MPV values in the control group and in the sufficient and the poor glycaemic control DM groups, p = 0,04 (One Way ANOVA, Fisher LSD).**

was associated with a higher risk of thromboembolism. Nevertheless, most of the patients from this cohort had type 2 diabetes [39].

In this study, we documented that conventionally used coagulation measurements are not adequately sensitive to detect moderate coagulation abnormalities, but using T-TAS system we are able to detect a significant difference in time of initiation of thrombus formation. Our results remained consistent with other research on hypercoagulability in Diabetes. Tripodi et al. revealed higher thrombin generation and an increased number of circulating microparticles in DM in comparison to the control cohort, whilst obtaining conventional coagulation

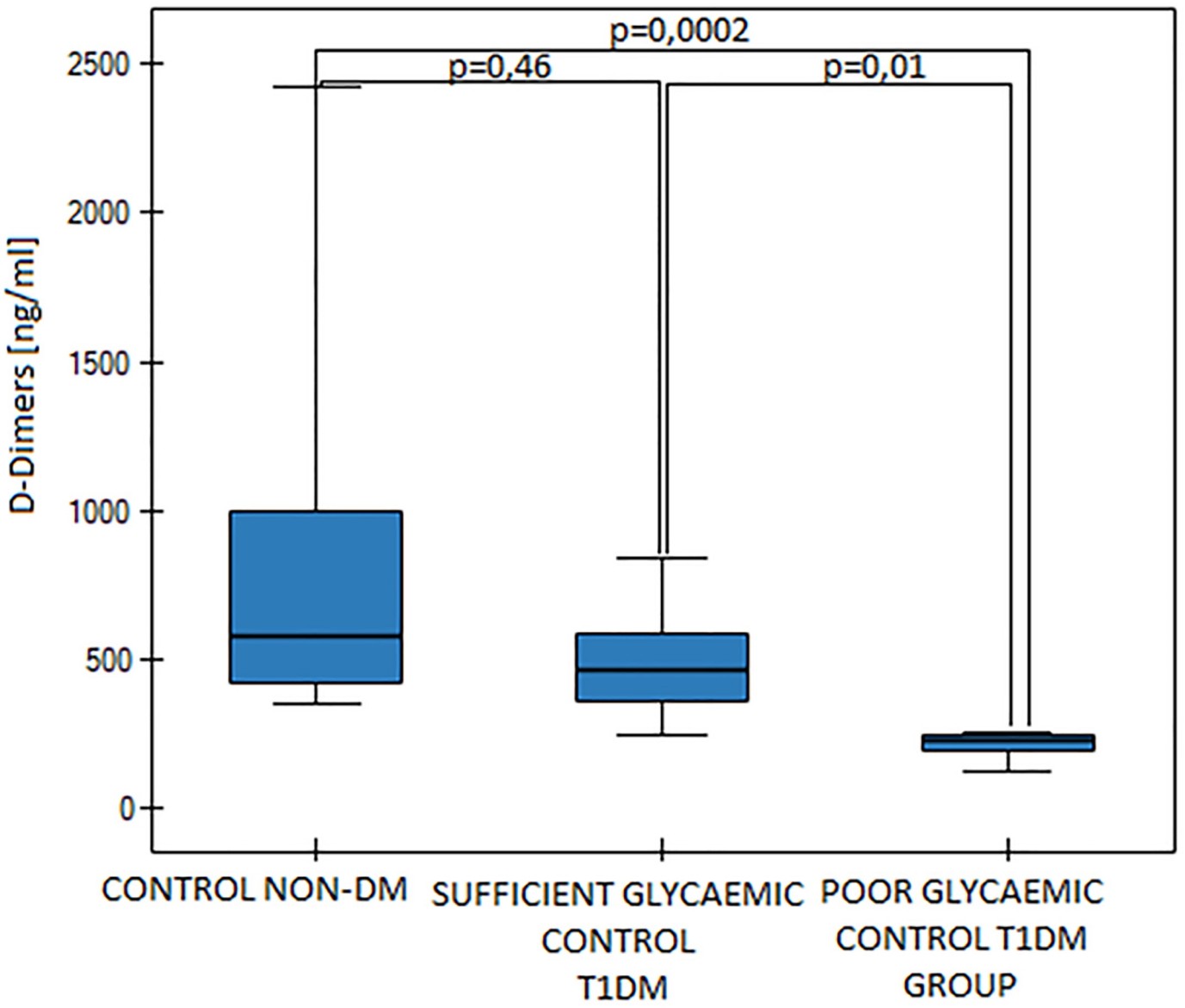

**Fig 3. D-dimers values in the control group and in the sufficient and the poor glycaemic control DM groups, p = 0,0004 (Kruskall-Wallis ANOVA, Dunn-Bonferroni).**

tests that difference was not detected [40]. The only difference between well and poorly controlled groups that was detected with standard coagulation measurements was a decreased D-dimers concentration, which might have been related to hypofibrinolysis.

In conclusion, our findings strongly support the thesis that poor glycemic control in DM subjects triggers a shift toward a pro-thrombotic and anti-fibrinolytic state. Results obtained by this system have shown that a relatively small increase in $Hb_{A1c}$ values in DM pregnant patients predisposes to the prothrombotic state during pregnancy, which cannot be detected using conventional coagulation tests. Nevertheless, it does appear to be detectable using a novel microchip flow-chamber system for quantitative assessment of the platelet thrombus formation process under flow conditions *in vitro*.

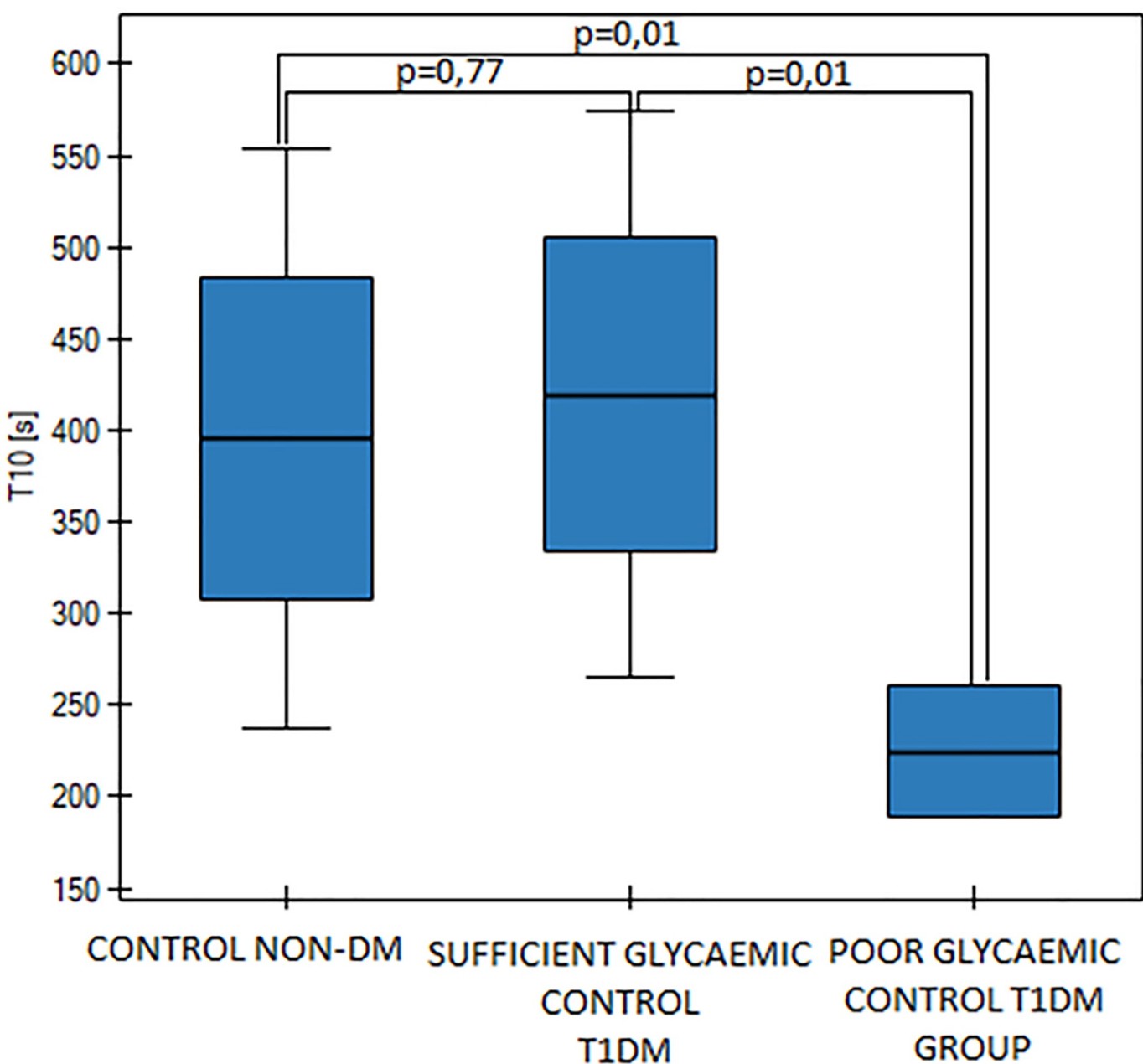

**Fig 4. T10 values in the control group and in the sufficient and the poor glycaemic control DM groups, p = 0,03 (One Way ANOVA, Fisher LSD).**

Another predictor of increased thrombogenicity in deficient glycemic control in pregnant women with Diabetes is elevated mean platelet volume values which correlates with T-TAS measurements. Thus MPV is an easily measurable parameter that might be helpful in predicting thrombotic state in diabetic pregnant women with poor glycemic control. Likewise, we observed a significant decrease in d-dimers concentration in the PGC DM group. Due to limited data and limited sample size of our study, the role of MPV, D-dimer and T-TAS alterations in poor glycemic control DM pregnant patients need to be further elucidated with more massive data sets in order to confirm our important findings.

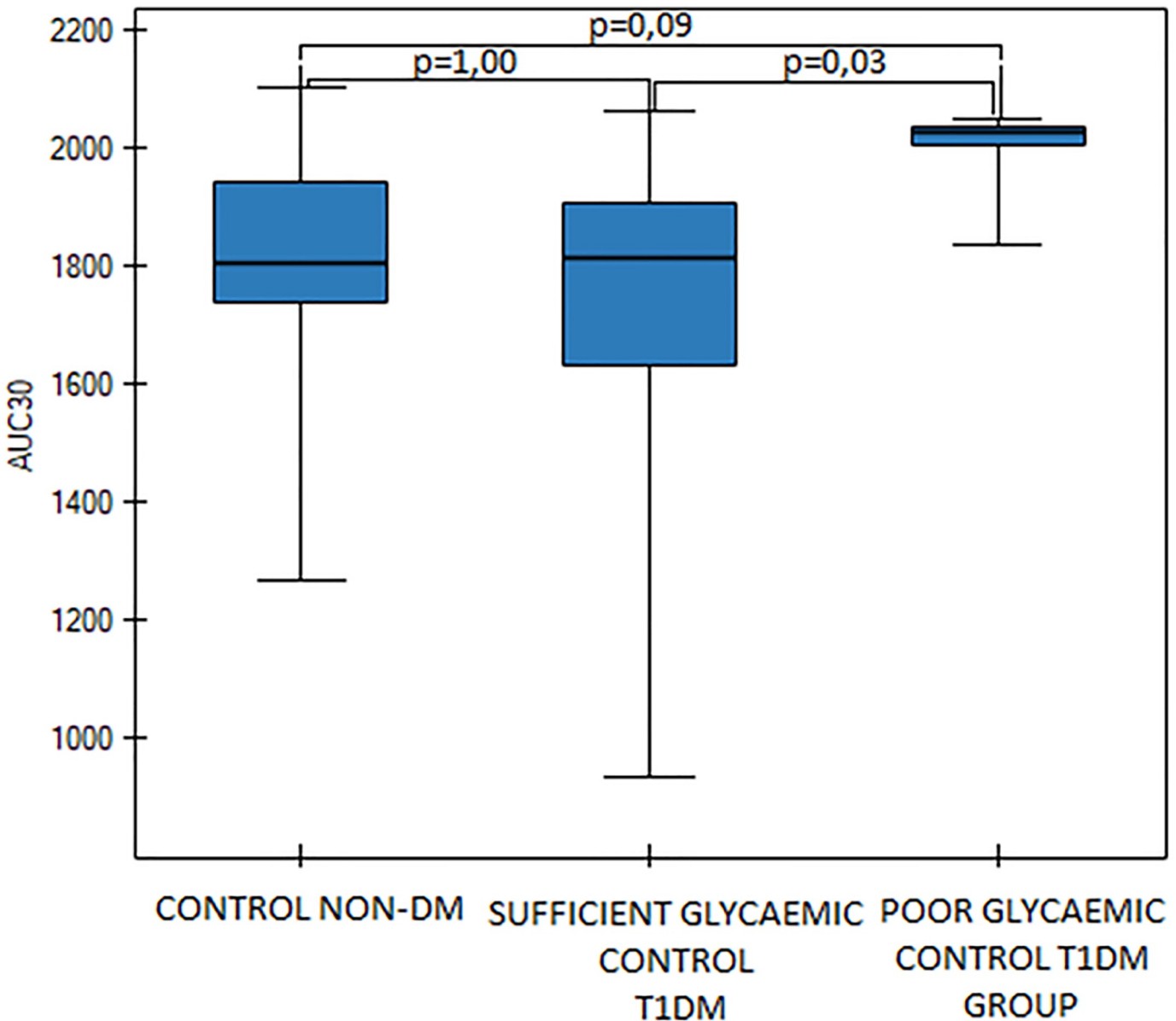

**Fig 5. AUC30 values in the control group and in the sufficient and the poor glycaemic control DM groups, p = 0,03 (Kruskall-Wallis ANOVA, Dunn-Bonferroni).**

## Supporting information

**S1 Table. Full characteristics of the studied groups.**
(XLSX)

## Author Contributions

**Conceptualization:** Maciej Osiński, Małgorzata Kędzia, Paweł Gutaj, Ewa Wender-Ożegowska.

**Data curation:** Maciej Osiński, Urszula Mantaj, Ewa Wender-Ożegowska.

**Formal analysis:** Maciej Osiński.

**Funding acquisition:** Maciej Osiński, Paweł Gutaj.

**Investigation:** Maciej Osiński, Małgorzata Kędzia, Ewa Wender-Ożegowska.

**Methodology:** Maciej Osiński, Ewa Wender-Ożegowska.

**Project administration:** Maciej Osiński.

**Resources:** Maciej Osiński, Małgorzata Kędzia, Paweł Gutaj.

**Software:** Maciej Osiński.

**Supervision:** Maciej Osiński, Urszula Mantaj, Małgorzata Kędzia, Ewa Wender-Ożegowska.

**Validation:** Maciej Osiński.

**Visualization:** Maciej Osiński, Małgorzata Kędzia, Paweł Gutaj.

**Writing – original draft:** Maciej Osiński.

**Writing – review & editing:** Maciej Osiński, Ewa Wender-Ożegowska.

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
