## [Decision Letter · Decision Letter 0]

22 Apr 2020

PONE-D-20-07710

Poor glycaemic control contributes to a shift towards prothrombotic and antifibrinolytic state in pregnant women with type 1 diabetes mellitus

PLOS ONE

Dear dr Osiński,

Thank you for submitting your manuscript to PLOS ONE. After careful consideration, we feel that it has merit but does not fully meet PLOS ONE’s publication criteria as it currently stands. Therefore, we invite you to submit a revised version of the manuscript that addresses the points raised during the review process.

We would appreciate receiving your revised manuscript by Jun 06 2020 11:59PM. To enhance the reproducibility of your results, we recommend that if applicable you deposit your laboratory protocols in protocols.io, where a protocol can be assigned its own identifier (DOI) such that it can be cited independently in the future. For instructions see: http://journals.plos.org/plosone/s/submission-guidelines#loc-laboratory-protocols

We look forward to receiving your revised manuscript.

Kind regards,

Christoph E Hagemeyer, PhD

Academic Editor

PLOS ONE

Journal Requirements:

2. In your Methods section, please provide additional information about the participant recruitment method and the demographic details of your participants. Please ensure you have provided sufficient details to replicate the analyses such as: a) a description of any inclusion/exclusion criteria that were applied to participant recruitment, b) a table of relevant demographic details, c) a statement as to whether your sample can be considered representative of a larger population.

3. Please include your tables as part of your main manuscript and remove the individual files. Please note that supplementary tables (should remain/ be uploaded) as separate "supporting information" files

Additional Editor Comments (if provided):

Reviewers' comments:

Reviewer's Responses to Questions

**Comments to the Author**

1. Is the manuscript technically sound, and do the data support the conclusions?

Reviewer #1: Yes

2. Has the statistical analysis been performed appropriately and rigorously? 

Reviewer #1: Yes

3. Have the authors made all data underlying the findings in their manuscript fully available?

Reviewer #1: Yes

4. Is the manuscript presented in an intelligible fashion and written in standard English?

Reviewer #1: Yes

5. Review Comments to the Author

Reviewer #1: The authors describe the use of the T-TAS system to evaluate the thombotic state of preganant women with and without diabetes. This study is designed well and conducted rigorously. There are a few minor commments to make:

1. please describe the term T80-10 in the methods.

2. Is T10 the time it takes to reach 10% of maximum pressure or is it defined different? Please add it to the methods.

3. Figure 4 needs representative traces of the actual T-TAS system to inform the reader of the shape and progression of the curves, particularly because the OT and T80-10 is not significantly different between groups.

4. The manuscript would benefit from an extra round of proof reading since it contains several grammatical errors throughout.

6. PLOS authors have the option to publish the peer review history of their article (what does this mean?). If published, this will include your full peer review and any attached files.

Reviewer #1: No

---

## [Author Response · Author response to Decision Letter 0]

8 Jul 2020

Dear Reviewer,

We are very grateful for your review. We would like to thank you for a very valuable and matter-of-fact review and for your suggestions of a minor comments. We believe we had addressed very carefully every one of them. 

Ad.1 (Please describe the term T80-T10 in the methods)

Response: please find attached file with a revised manuscript track changes. From the revised manuscript: “T80-T10 – time of growth of the thrombus on the AR chip, (time between time of blood clot formation initiation and occlusion time). And more later: “T80-T10 is the interval between T10 and OT or T80 and represents the rate of growth of a thrombus.”

Ad. 2. (Is T1 Is T10 the time it takes to reach 10% of maximum pressure or is it defined different? Please add it to the methods.)

Response: please find attached file with a revised manuscript track changes. From the revised manuscript: “ T10 is defined as the time of the onset of a thrombus formation. It resembles the duration of the flow pressure from baseline values to the 10 kPa pressure values. . It is related with a partial occlusion of microcapillary. ”

Ad. 3. (Figure 4 needs representative traces of the actual T-TAS system to inform the reader of the shape and progression of the curves, particularly because the OT and T80-10 is not significantly different between groups.)

Response: 

In our paper we showed that T-TAS® results substantially differed between the group with 

sufficient and the group with poor glycemic control. Blood clot formation initiation time was significantly shortened in poorly controlled type 1 diabetic pregnant patients.

There have been numerous scientific data with T-TAS® usage. However in those studies patients were suffering from a heart diseases, were older and had different metabolic diseases. Basically there has been no such research in a pregnant women group at all do date, and moreover no any data in Type 1 Diabetic pregnant patients. We find that notion both innovation and an incentive to a further haemostasis investigation in poor controlled Type 1 Diabetic pregnant women on a larger groups of patients. Moreover we would like to articulate in scientific press a notion of altered whole blood thrombogenicity in poor controlled diabetic patients and thus encourage other researchers to carry out a similar research, not only in a pregnant patients but in the other groups of patients suffering from this disease. 

So unfortunately, due to above-mentioned notions, we cannot add any other representative traces of actual TTAS system in pregnancy to fig. No 4 , because our research is the only one on this subject to date; and the only traces that can be compared with a study (diabetes) groups is our healthy pregnant well-matched control group. We hope our research paper will be a stimulus to a further investigations of a whole blood thrombogenicity in a pregnant women. We also highly believe that this phenomenon demands a further research, while pregnancy significantly increases risk of developing an episode of arterial and, especially, venous thrombosis. 

Ad. 4. (The manuscript would benefit from an extra round of proof reading since it contains several grammatical errors throughout.)

Response: extra round of proof reading had been carefully performed, and errors had been corrected throughout.

Yours sincerely,

Authors

---

## [Editor Report · Decision Letter 1]

5 Aug 2020

Poor glycaemic control contributes to a shift towards prothrombotic and antifibrinolytic state in pregnant women with type 1 diabetes mellitus

PONE-D-20-07710R1

Dear Dr. Osiński,

We’re pleased to inform you that your manuscript has been judged scientifically suitable for publication and will be formally accepted for publication once it meets all outstanding technical requirements.

Kind regards,

Christoph E Hagemeyer, PhD

Academic Editor

PLOS ONE
---

## [Editor Report · Acceptance letter]

7 Aug 2020

PONE-D-20-07710R1 

Poor glycaemic control contributes to a shift towards prothrombotic and antifibrinolytic state in pregnant women with type 1 diabetes mellitus 

Dear Dr. Osiński:

I'm pleased to inform you that your manuscript has been deemed suitable for publication in PLOS ONE. Congratulations! Your manuscript is now with our production department. 

Kind regards, 

on behalf of

Dr. Christoph E Hagemeyer 

Academic Editor

PLOS ONE